# Protein-mediated RNA folding governs sequence-specific interactions between rotavirus genome segments

**Alexander Borodavka[1,2]\*, Eric C Dykeman[3,4,5], Waldemar Schrimpf[2], Don C Lamb[2]**

[1]Astbury Centre for Structural Molecular Biology, School of Molecular and Cellular Biology, University of Leeds, Leeds, United Kingdom; [2]Department of Chemistry, Center for NanoScience, Nanosystems Initiative Munich (NIM) and Center for Integrated Protein Science Munich (CiPSM), Ludwig-Maximilian University of Munich, Munich, Germany; [3]York Centre for Complex Systems Analysis, University of York, York, United Kingdom; [4]Department of Mathematics, University of York, York, United Kingdom; [5]Department of Biology, University of York, York, United Kingdom

**Abstract** Segmented RNA viruses are ubiquitous pathogens, which include influenza viruses and rotaviruses. A major challenge in understanding their assembly is the combinatorial problem of a non-random selection of a full genomic set of distinct RNAs. This process involves complex RNA-RNA and protein-RNA interactions, which are often obscured by non-specific binding at concentrations approaching in vivo assembly conditions. Here, we present direct experimental evidence of sequence-specific inter-segment interactions between rotavirus RNAs, taking place in a complex RNA- and protein-rich milieu. We show that binding of the rotavirus-encoded non-structural protein NSP2 to viral ssRNAs results in the remodeling of RNA, which is conducive to formation of stable inter-segment contacts. To identify the sites of these interactions, we have developed an RNA-RNA SELEX approach for mapping the sequences involved in inter-segment base-pairing. Our findings elucidate the molecular basis underlying inter-segment interactions in rotaviruses, paving the way for delineating similar RNA-RNA interactions that govern assembly of other segmented RNA viruses.

DOI: https://doi.org/10.7554/eLife.27453.001

**\*For correspondence:**
A.Borodavka@leeds.ac.uk

**Competing interests:** The authors declare that no competing interests exist.

## Introduction

Genomes of rotaviruses (RVs), and other pathogens of the *Reoviridae* family, comprise nine to twelve double-stranded (ds)RNA segments, co-packaged into each infectious virion. The RNA assortment process, during which a single distinct positive-sense ssRNA copy of each of the genomic segments is selected for packaging and replication, occurs within cytoplasmic inclusion bodies termed viroplasms (*Patton and Spencer, 2000*; *Trask et al., 2012*). Multiple copies of viral RNAs and proteins accumulate in viroplasms, of which the non-structural ssRNA-binding proteins NSP2 & NSP5 are essential components (*Patton and Spencer, 2000*; *Trask et al., 2012*; *Taraporewala and Patton, 2004*). Current views of the mechanisms of segment assortment are based on the idea that selection of the correct genomic segments is governed by inter-segment RNA-RNA interactions (*Gavazzi et al., 2013a*; *Gavazzi et al., 2013b*; *Fajardo et al., 2015*; *McDonald et al., 2016*). However, the analysis of such interactions remains a particularly challenging task due to a vast number of RNA-RNA contacts present in multiple folding intermediates of large RNAs (*Solomatin et al., 2010*; *Woodson, 2010a*; *Woodson, 2010b*; *Laing and Schlick, 2011*). This problem is further confounded by non-specific RNA self-association and aggregation, particularly in the presence of RNA-binding

**eLife digest** Rotavirus is a highly infectious virus that affects children worldwide, causing severe diarrhoea. Despite the introduction of several highly effective vaccines, more than 200,000 children still die from rotavirus each year. There are currently no drugs that can combat this disease once a child has been infected.

Viruses carry the instructions that determine their properties and behavior in molecules of DNA or RNA. Unlike many other viruses, which typically have a single molecule of DNA or RNA, rotavirus has 11 distinct "RNA segments". After invading a cell the virus begins to replicate itself. During replication, the RNA segments (which consist of two strands of RNA paired together) are copied many times. It is not clear how rotaviruses 'count' up to 11 so that each new virus acquires a single copy of each segment.

Previous biochemical and structural studies of rotavirus replication suggest that selecting 11 distinct RNA segments must involve the RNAs forming complex interactions with proteins and other RNA molecules. Using a highly sensitive fluorescence-based approach, termed fluorescence cross-correlation spectroscopy, Borodavka et al. now present direct experimental evidence of interactions between the RNA segments that occur via single strands of the rotavirus RNA.

These RNA-RNA interactions require the binding of a rotavirus protein NSP2 to the RNA strands, which results in the remodeling of the RNA; this remodeling is required to form stable contacts between different RNA segments. Furthermore, a new experimental approach (called RNA-RNA SELEX) developed by Borodavka et al. identified the parts of the RNA segments that may take part in these interactions.

The results presented by Borodavka et al. pave the way for identifying the RNA-RNA interactions that govern how other segmented RNA viruses can package their genetic material. Further work to uncover the entire RNA interaction network in rotaviruses would also accelerate the design of new vaccines and may help us to develop antiviral drugs to treat infections.
DOI: https://doi.org/10.7554/eLife.27453.002

proteins, at concentrations mimicking in vivo assembly conditions (*Borodavka et al., 2012*). Recently, several tools for investigating RNA-RNA interactions have been developed, based on proximity-ligation of interacting RNAs (*Helwak and Tollervey, 2014*) and psoralen-mediated crosslinking of RNA duplexes (*Engreitz et al., 2014*; *Lu et al., 2016*; *Sharma et al., 2016*; *Aw et al., 2016*). While these powerful methods are well suited for detection of RNA duplexes in cells, further experimental validation of the identified RNA-RNA contacts is often required (*Weidmann et al., 2016*). Moreover, such techniques give little insight into the dynamics and stability of the observed RNA-RNA interactions, particularly when they are involved in the assembly of macromolecular complexes.

To address these challenges, we have developed an experimental framework for interrogating RNA-RNA interactions by taking advantage of two-colour fluorescence cross-correlation spectroscopy combined with pulsed interleaved excitation (PIE-FCCS) (*Müller et al., 2005*; *Schrimpf et al., 2017*). Such assays offer unprecedented capacity for detection of stable, sequence-specific interactions between labeled RNAs in complex mixtures of RNAs and proteins. Using PIE-FCCS, we show that incubation of a full set of eleven genomic ssRNAs with the non-structural protein NSP2 results in de novo formation of specific inter-segment RNA-RNA contacts. By testing pairwise RNA-RNA interactions between segment 11 ssRNA (S11) and other RV ssRNAs, we demonstrate that S11 preferentially binds to a subset of genomic ssRNAs. We show that NSP2 binding to S11 RNA results in its structural reorganization, concomitant with the exposure of single-stranded areas, required for stabilization of new inter-segment interactions. The weakest detected pairwise interaction between S11 and S10 RNAs is significantly enhanced in the presence of a full set of eleven ssRNAs, suggesting formation of a complex RNA interaction network, stabilized by NSP2.

To gain further insights into specific inter-segment contacts of RV RNAs, we introduce an RNA-RNA SELEX approach for mapping the genomic sequences, mediating such interactions. Using FCCS methodology combined with RNA mutagenesis studies, we validate the sites of inter-segment RNA-RNA interactions, identified via RNA-RNA SELEX. This integrated approach provides unique

insights into the stability of macromolecular complexes, containing multiple RNAs, and it can be universally applied for investigating assembly of other segmented RNA viruses and ribonucleoproteins.

## Results

### NSP2 promotes interactions between ssRNA segment precursors

Unlike most ensemble methods previously used for detecting inter-segment interactions in segmented RNA viruses (*Gavazzi et al., 2013b*; *Fajardo et al., 2015*; *Fournier et al., 2012*), fluorescence correlation spectroscopy (FCS) allows probing of such interactions in extremely dilute solutions, effectively eliminating self-association and aggregation of RNAs (*Borodavka et al., 2012*; *Borodavka et al., 2016*). We employed this technique to identify specific inter-molecular RNA-RNA interactions, which remain stable at low sub-nanomolar concentrations. We used a dual-color extension of FCS, fluorescence cross-correlation spectroscopy (FCCS, Materials and methods and *Figure 1—figure supplement 1*) for detecting interactions of differently labeled RNAs in the presence of unlabeled molecules, which may be required for stabilization of such RNA-RNA contacts.

We first investigated whether the RV ssRNAs could spontaneously associate into larger RNA complexes in vitro. We examined binding of S1-S10 ssRNAs (*Figure 1—figure supplement 2*) to the smallest genomic RNA S11, for which there is a complete secondary structure model available (*Li et al., 2010*). As described in Materials and methods, ATTO647-labeled S11 and ATTO565-labeled S10 (+)ssRNAs were incubated with unlabeled (+)ssRNAs S1 to S9. After incubation, RNA samples were diluted (1 nM each) and examined by two-color FCCS. Analysis of the cross-correlation function (CCF) between dye-labeled S11 and S10 RNAs suggests that the two RNAs do not interact with each other in the presence of a full set of eleven genomic ssRNA segment precursors (*Figure 1A*, zero amplitude of CCF, shown as dashed magenta line). Upon incubation with eleven genomic ssRNAs, the apparent hydrodynamic radius, $R_h$ of the S11 RNA remained unchanged, consistent with lack of RNA oligomerization under those conditions (*Supplementary file 1*).

Because RNA segment assortment is likely to occur in the presence of the non-structural proteins NSP5 and NSP2 (3, 24), the latter capable of helix-unwinding and strand-annealing reactions in vitro (*Taraporewala and Patton, 2001*; *Borodavka et al., 2015*), we first examined inter-segment RNA interactions in the presence of NSP2. Incubation of eleven distinct ssRNAs S1-S11 with a 100-fold molar excess of NSP2, that is, enough NSP2 to promote strand-annealing (*Borodavka et al., 2015*), both labeled S11 and S10 formed a stable complex (*Figure 1A*, CCF in blue).

Analysis of the resulting ACFs and the CCF confirms that both NSP2-bound RNAs have similar hydrodynamic radii, $R_h$ ~30 nm, significantly larger than the protein-free S11 and S10 ssRNAs (*Supplementary file 1*).

Having established that the S11:S10 interaction is formed in the presence of a full genomic set of eleven ssRNAs upon incubation with NSP2, we then explored whether the remaining S1-S9 RNAs are involved in stabilization of the detected interaction. Omitting S1-S9 RNAs from the reaction significantly decreased the apparent CCF amplitude between S11 and S10 RNAs (*Figure 1*, *cf*. panels A and B). This result suggests that the observed S11:S10 interaction involves additional RNAs that may be required to form a larger supramolecular complex (*Trask et al., 2012*; *Gavazzi et al., 2013a*; *Gavazzi et al., 2013b*). Alternatively, binding of one or more RNAs to either S11 or S10 may result in stabilization of the RNA conformations, which favor stable base-pairing between the two RNAs. Either interaction model predicts that some of the remaining segment ssRNAs (S1-S9) should strongly interact with S11 RNA.

We then examined the remaining pairwise interactions between S11 and S1-S9 RNAs, after incubating an equimolar mixture of the two differently labeled RNAs with NSP2 (*Figure 1B*). The strongest pairwise interactions of S11 were observed for segments S3, S6 and S5, with CCF amplitudes similar to those, observed for the S11:S10 complex, incubated in the presence of S1-S9 RNAs (CCFs ~ 0.5, *Figure 1A*). The apparent CCF amplitudes between S11 and any other segment RNA are not proportional to the length or the size of the interacting RNA partner (*Supplementary file 1*), suggesting that it is the RNA sequence, rather than its size, that is important for S11 binding.

We also examined the effect of NSP5 on ssRNA binding and inter-segment RNA complex formation. Both NSP2 and NSP5 were capable of binding a short 18-nt long ssRNA with nM affinity, yielding the apparent hydrodynamic radii, R$h$, closely matching the sizes of NSP2 octamers

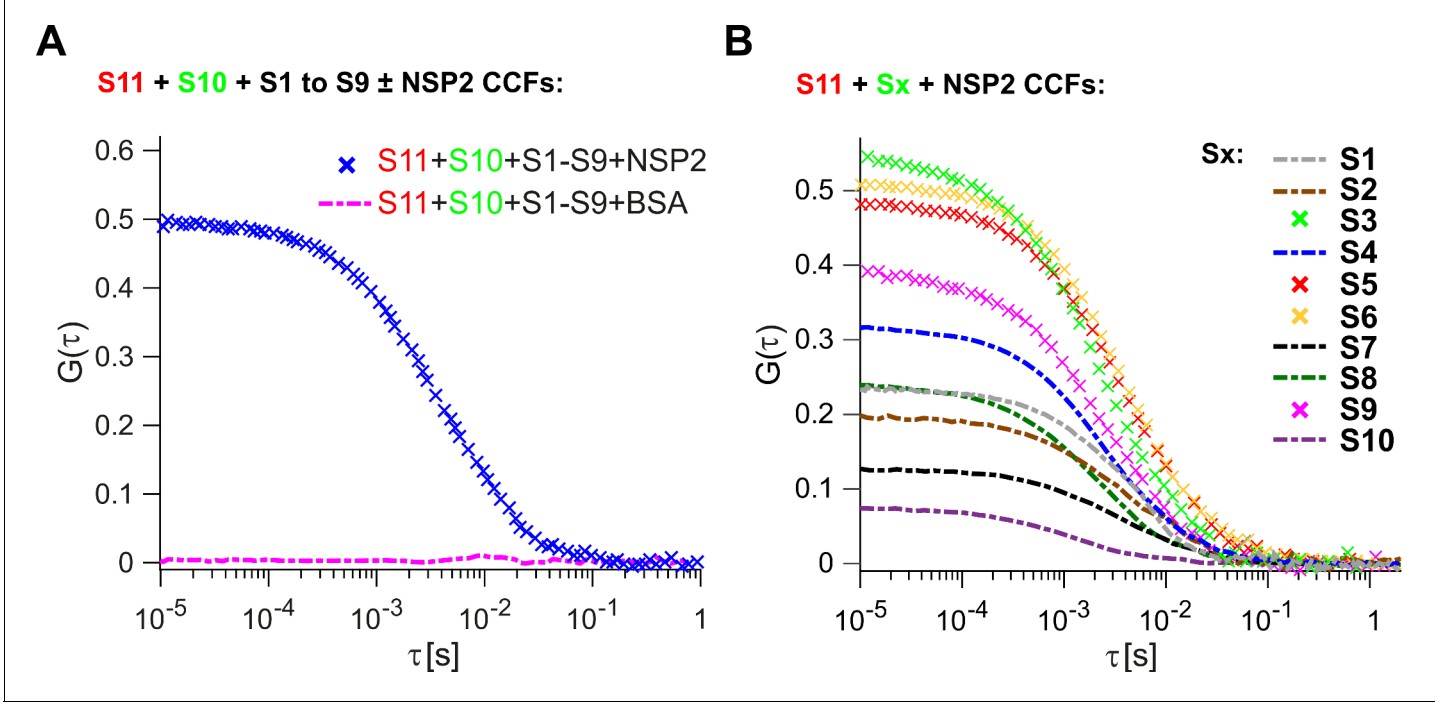

**Figure 1.** Inter-segment RNA-RNA interactions in rotaviruses examined by FCCS. (**A**) Interactions between fluorescently labelled ssRNAs S10 and S11, incubated in the presence of an equimolar set of unlabelled ssRNAs S1 to S9, were probed by FCCS, as described in Materials and methods. A full set of S1-S11 ssRNAs does not associate into a higher order RNA complex (cross-correlation function amplitude, CCF ≈ 0, shown as a dashed magenta line). In contrast, incubation of the mixture of all eleven genomic ssRNAs in the presence of NSP2 results in a strong interaction between S10 and S11 (high amplitude of CCF shown in blue). (**B**) Pairwise inter-segment RNA-RNA interactions in RVs. Fluorescently labelled ssRNA S11 (red), incubated with RNA Sx (where x – any other genomic ssRNA S1-S10) in the presence of NSP2, as described in Materials and methods. CCF amplitudes were normalised by their respective auto-correlation functions (ACFs), reflecting the differences in the affinities between S11 and other ssRNAs, with the strongest interactions detected for the RNA-RNA combinations S11:S3, S11:S5 and S11:S6.

DOI: https://doi.org/10.7554/eLife.27453.003

The following source data and figure supplements are available for figure 1:

**Source data 1.** Normalised CCFs of the non-interacting fluorescently labelled ssRNAs S10 and S11, incubated with BSA in the presence of an equimolar set of unlabelled ssRNAs S1 to S9 (CCF = 0), and the CCFs of the interacting S10 and S11 co-incubated with NSP2.

DOI: https://doi.org/10.7554/eLife.27453.010

**Source data 2.** Normalised CCFs of the interacting RNAs S11 and Sx (x – any other RNA S1-S10) in the presence of NSP2.

DOI: https://doi.org/10.7554/eLife.27453.011

**Figure supplement 1.** Principles of Dual-color Fluorescence Cross-Correlation Spectroscopy (FCCS).

DOI: https://doi.org/10.7554/eLife.27453.004

**Figure supplement 2.** RV genomic segment ssRNAs used in the study.

DOI: https://doi.org/10.7554/eLife.27453.005

**Figure supplement 3.** RNA-RNA interactions examined by FCCS in the presence of NSP2 and NSP5.

DOI: https://doi.org/10.7554/eLife.27453.006

**Figure supplement 4.** Effect of ATP on the RNA-binding properties of NSP2.

DOI: https://doi.org/10.7554/eLife.27453.007

**Figure supplement 5.** S5:S11 RNA-RNA interactions examined in the presence of RV group A NSP2 mutant ΔC-NSP2 and RV group C NSP2.

DOI: https://doi.org/10.7554/eLife.27453.008

**Figure supplement 6.** Interactions of ssRNAs S5 and S11 after proteolytic removal of NSP2.

DOI: https://doi.org/10.7554/eLife.27453.009

(*Schuck et al., 2001*), or NSP5 decamers (*Martin et al., 2011*), respectively (*Figure 1—figure supplement 3A*). Despite its ssRNA-binding activity, incubation of NSP5 with ssRNAs S5 and S11 at saturating amounts did not yield any detectable cross-correlation (*Figure 1—figure supplement 3B*), indicating that NSP5 binding to ssRNAs did not promote the formation of a stable S5:S11 RNA complex. Remarkably, co-incubation of NSP2 in the presence of sub-micromolar amounts of NSP5

resulted in a marked decrease of the S5:S11 CCF amplitude, compared to the NSP2 incubation alone (*Figure 1—figure supplement 3C*). Further addition of NSP5 resulted in severe aggregation of NSP2, as previously reported by *Jiang et al. (2006)*.

Given the NTPase and autophosphorylation activities of NSP2 (30–33), we also investigated the effect of ATP on the formation of RNA-RNA contacts in the presence of NSP2. Addition of 1 mM ATP did not affect RNA binding by NSP2 octamers (*Figure 1—figure supplement 4A*), nor had it any effect on the apparent S11:S5 CCF amplitude (*Figure 1—figure supplement 4B*). Collectively these data suggest that inter-segment RNA contact formation is not coupled to ATP hydrolysis and thus most likely to be dependent on the ssRNA-binding activity of NSP2. To further investigate this, we examined the interactions between S5 and S11 RNAs co-incubated with a C-terminally truncated NSP2 mutant (ΔC-NSP2) with a significantly reduced affinity for ssRNA (*Hu et al., 2012*). The analysis of S11 and S5 RNAs ACFs reveals that both interacting RNAs recruited multiple copies of the full-length NSP2 or its deletion mutant ΔC-NSP2, respectively, yielding similar hydrodynamic sizes of S5 (*Figure 1—figure supplement 5A*, light and dark green ACFs). However, the apparent CCF amplitude between S5 and S11 was significantly reduced when the full-length NSP2 was replaced with ΔC-NSP2 (*Figure 1—figure supplement 5A*, CCFs in blue and magenta). A drastic reduction of the CCF amplitude between S5 and S11 ssRNAs was also observed when rotavirus group A NSP2 was substituted with NSP2 from rotavirus group C (*Figure 1—figure supplement 5B*), despite the apparent structural resemblance and similar enzymatic activities of both proteins (*Hu et al., 2012*). Together, these results strongly suggest that the formation of stable inter-segment RNA-RNA contacts requires selective binding of RV group A NSP2 to ssRNAs, but not ssRNA-binding proteins NSP5 or RV group C NSP2.

## Inter-segment interactions are mediated by direct RNA-RNA contacts

Given that NSP2 has a capacity to bind multiple RNAs (*Schuck et al., 2001*), the observed cross-correlation may reflect its simultaneous binding to differently labeled segment ssRNAs. The differences in relative affinities between S11 and other ssRNAs (*Figure 1B*) strongly suggest that the detected RNA interactions are not due to non-specific aggregation, resulting from NSP2 binding. Moreover, the observed inter-segment interactions remained stable in highly dilute solutions (<1 nM), while the apparent affinity of NSP2 for ssRNA is nanomolar (*Hu et al., 2012*). Thus, we hypothesized that the high stability of these interactions could arise from strong, specific inter-molecular RNA contacts, formed upon incubation with NSP2. To test this, we removed NSP2 from the preformed stable RNA-RNA complex S11:S5 by digesting it with proteinase K, and examined the sample by FCCS. The interacting S5 and S11 RNAs had lower $R_h$ upon removal of NSP2 (*Table 1* and *Figure 1—figure supplement 6*), suggesting that both RNAs were protein-free. These highly diluted NSP2-free RNA-RNA complexes remained stable at 37°C for extended periods of time (>30 min, *Figure 1—figure supplement 6A*).

As a further control, we made a protein-free RNA-RNA complex S11:S5 by heat-annealing a mixture of fluorescently labeled RNAs S11 and S5 (Materials and methods), and examined it by FCCS. Both heat-annealed RNAs strongly interacted (*Figure 1—figure supplement 6B*), yielding an apparent $R_h$ ~13–17 nm typical for a proteinase K-digested complex S11:S5 (*Table 1*). Together, these data strongly support the view that inter-segment interactions mediated by binding of NSP2 remain stable after removal of NSP2 due to the formation of new, specific inter-molecular RNA contacts.

**Table 1.** Apparent hydrodynamic radii ($R_h$) of the protein-free and NSP2-bound genomic ssRNAs

|  | Rh of S11 | Rh of S5 |
| --- | --- | --- |
| S11 + S5, no NSP2 | 9 ± 0.9 nm | 13.4 ± 0.95 nm |
| S11 + S5, heat-annealed | 13.3 ± 1.3 nm | 17.0 ± 3.0 nm |
| S11 + S5+NSP2 | 17.3 ± 2.9 nm | 20.8 ± 5.0 nm |
| S11 + S5+NSP2, proteinase K digested | 13.0 ± 1.4 | 17.0 ± 2.3 nm |

The $R_h$ values are reported as mean ±SD computed from at least 6 measurements.

DOI: https://doi.org/10.7554/eLife.27453.012

## Exploring the S11 RNA interactome using RNA-RNA SELEX

Having established that S11 preferentially binds to S3, S5 and S6 RNAs, we devised an RNA-RNA SELEX methodology to identify the S11 RNA sequences that can form stable inter-molecular contacts with other segment ssRNAs. We used this approach to analyze the distribution of the RNA-RNA interaction sites in S11 alone, and in complex with NSP2 (*Figure 2A*).

As described in Materials and methods, we used a naïve RNA library comprising ~3.6×10¹³ unique 30-mer ssRNA sequences. Biotinylated S11 RNA was incubated with the naïve RNA library (*Figure 2A*), and sequences strongly interacting with immobilized S11 RNA were separated from excess NSP2 and low affinity 30-mers through multiple rounds of washes. After several rounds of sequence enrichment, the resulting pool of 30-mers was subjected to high-throughput sequencing, yielding $4.9 \times 10^6$ unique sequences. To identify the sequences that strongly interact with S11 RNA, we aligned the reverse complements of all individual SELEX-enriched sequences to S11 using a probabilistic score-based approach (Materials and methods). The resulting histogram indicates the areas, in which multiple SELEX-enriched sequences align to S11 RNA (the peaks shown as a black dashed line, *Figure 2B*). The histogram peaks are correlated with highly exposed areas of S11 RNA that strongly interact with ssRNAs during the RNA-RNA SELEX procedure. Similarly, S11 areas with poor accessibilities, e.g., stable helices, yielded the lowest number of sequence alignments.

As the test of the ability of our approach to report on secondary structure of S11 RNA, we computed its structure using the constraints, derived from the RNA-RNA SELEX experiment (Materials and methods). The analysis enabled us to precisely map the conserved helices H1-H3 (*Figure 2B* and *Figure 2—figure supplement 1*), consistent with the structure probing data and covariation analyses (*Li et al., 2010*).

Similarly, multiple histogram peaks aligned well to various loop regions that had been previously identified in S11 RNA (*Figure 2—figure supplement 1*). Overall, the RNA-RNA SELEX data reproduced multiple features of the target RNA structure, demonstrating that our approach does not perturb its solution conformations. More importantly, RNA-RNA SELEX allows direct identification of the RNA areas capable of forming stable inter-molecular contacts.

## Revealing NSP2-mediated conformational changes in S11 RNA

Because the folded structures of interacting RNAs in isolation would present a substantial energy barrier for forming inter-molecular base pairs between them, RNA-RNA SELEX favors selection of unstructured RNA sequences. We reasoned that binding of NSP2 to ssRNAs would alter their structures and, thus, accessibilities of the RNA-binding sites. Analysis of the RNA-RNA SELEX data revealed significant enrichment of the RNAs interacting with S11 in the presence of NSP2 (*Figure 2B*, solid red line), consistent with its strand-annealing activity (*Taraporewala and Patton, 2001*; *Borodavka et al., 2015*). A comparison between the two SELEX experiments uncovered a number of additional RNA-binding sites in S11 RNA after its incubation with NSP2 (*Figure 2B* and *Figure 2—figure supplement 2A*). The overall architecture of S11, including stable helices H1-H3, was not affected by NSP2 binding (*Figure 2B and C*). Interestingly, the majority of the newly formed RNA-binding sites were located near the loop regions and helical junctions of the RNA (*Figure 2C*), potentially reflecting the sites of preferential binding of NSP2 to S11 RNA.

To further confirm that NSP2 binding to S11 results in conformational change of the RNA, we monitored the circular dichroism (CD) spectrum of S11 RNA in the presence of NSP2, as described in Materials and methods. The protein-free S11 RNA spectrum is characteristic of an A-type helical conformation (*Figure 2—figure supplement 2B*), typical for a folded RNA (*Borodavka et al., 2015*). A positive 260–265 nm band decreases in response to increasing amounts of NSP2 (*Figure 2—figure supplement 2B*, shown in blue), or thermal melting (shown in red), indicating that the global helical RNA fold is destabilized by NSP2 binding, consistent with the structural rearrangements due to NSP2-mediated RNA unfolding.

## Identification of sites of Inter-segment RNA-RNA interactions

Having established that NSP2 binding is a prerequisite for inter-segment RNA-RNA interactions, we aligned the SELEX-enriched sequences to the S1-S10 RNAs (*Figure 3*). The analysis revealed multiple sequences within S1-S10 genome segments with a potential to interact with S11 RNA (*Figure 3*, black peaks).

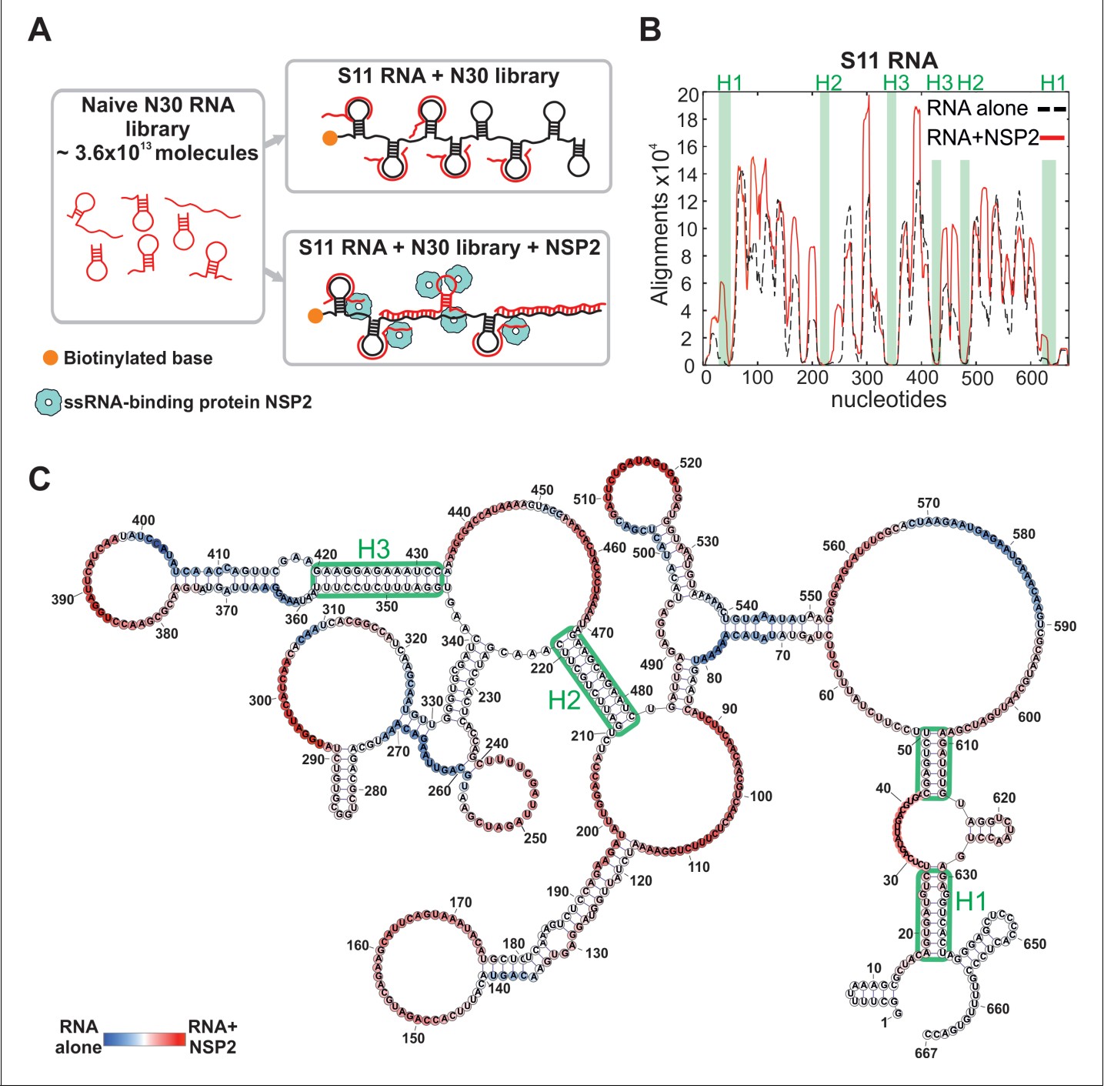

**Figure 2.** An RNA-RNA SELEX approach for mapping inter-segment interactions. (**A**) A naïve 30-nt ssRNA library contains a subset of RNAs (shown as red stem-loops) that can interact with the target RNA (schematically shown in black). In parallel, the naïve RNA library is incubated with S11 RNA in the presence of the RNA-binding protein NSP2 (schematically shown as blue spheres). Both RNA libraries are subject to an RNA-RNA SELEX procedure, combined with high-throughput sequencing, as described in Materials and methods. Analysis of the recovered sequences allows identification of multiple RNA-binding sites present in S11 RNA. (**B**) Histogram plots of the sequences recovered from the RNA-RNA SELEX against S11 RNA, which were aligned to the sequence of S11 (Materials and methods). The numbers of sequences aligning to a given nucleotide of S11 are normalised to the overall number of alignments in both RNA libraries (S11 RNA alone – black dashed line, S11 +NSP2 – solid red line). Highlighted in green are regions with low accessibilities that correspond to stable helices H1-H3, yielding poor sequence enrichment during the RNA-RNA SELEX procedure. (**C**) NSP2-induced structural rearrangements of S11 RNA. The refolded RNA structure was computed using folding constraints derived from the RNA-RNA SELEX

*Figure 2 continued on next page*

Figure 2 continued

experiment. The colour map shows changes in accessibilities in S11 RNA upon NSP2 binding. Highly exposed areas of the S11 RNA alone are shown in blue, while the sequences predominantly enriched in the presence of NSP2 are shown in red.

DOI: https://doi.org/10.7554/eLife.27453.013

The following source data and figure supplements are available for figure 2:

**Source data 1.** The number of sequences recovered from the RNA-RNA SELEX against S11 RNA alone, which were aligned to the sequence of S11 RNA and filtered using Bernouli score of 12.

DOI: https://doi.org/10.7554/eLife.27453.016

**Source data 2.** The number of sequences recovered from the RNA-RNA SELEX against S11 RNA in complex with NSP2, which were aligned to the sequence of S11 RNA and filtered using Bernoulli score of 12.

DOI: https://doi.org/10.7554/eLife.27453.017

**Source data 3.** A connectivity table (CT) file of the S11 RNA structure computed using folding constraints derived from the RNA-RNA SELEX experiment performed in the presence of NSP2.

DOI: https://doi.org/10.7554/eLife.27453.018

**Figure supplement 1.** Secondary structure model of S11 RNA.

DOI: https://doi.org/10.7554/eLife.27453.014

**Figure supplement 2.** Conformational changes in S11 RNA upon NSP2 binding.

DOI: https://doi.org/10.7554/eLife.27453.015

---

Remarkably, the distribution of the identified peaks changes in the presence of NSP2 (*Figure 3*, peaks shown in red), reflecting the observed conformational rearrangement of the S11 RNA.

The identified sequences appear in the RV RNA segments with high statistical significance (>200 hits using a statistical score of 14 or above for sequences, enriched in the presence of NSP2), compared to the non-rotaviral control sequence (Non-RV, Materials and methods and *Figure 3—figure supplement 1A*). This result suggests that the RV segment precursors may contain multiple sites that have the potential to interact with S11 RNA in the presence of NSP2.

Next, we compared the number of the hits resulting from RNA-RNA SELEX experiments, with the relative amplitudes of CCFs, determined in pairwise RNA-RNA interaction assays. There is a positive correlation between the abundance of S11-interacting sequences present in S1-S10 and the respective CCF amplitudes (*Figure 1B* and *Figure 3—figure supplement 1B*). Because RNA-RNA SELEX provides the information about the accessibility of a target RNA, but not its interacting partners, we analyzed the sequences of segments S3, S5 and S6, which strongly interacted with S11 in the presence of NSP2. We selected the peaks with significant sequence enrichment, capable of stable base-pairing with S11 RNA in the presence of NSP2 (*Figure 3*, peaks highlighted in magenta boxes). The identified sequences within S3, S5 and S6 RNAs can stably base-pair with S11 RNA ($\Delta G = -15.3$ to $-19.1$ kcal/mol, *Figure 3*, insets and *Figure 3—figure supplement 2*). Overall, the stabilities of the identified inter-segment RNA-RNA duplexes correlate well with the apparent CCF amplitudes.

We carried out similar analyses for RNAs S1, S4 and S10, which also contain sequences that have a potential of base pairing with S11 (*Figure 3*, highlighted in blue boxes). Despite the observed enrichment of the sequences interacting with S11 RNA, identified in S1, S4 and S10, the apparent CCF amplitudes determined in pairwise RNA-RNA interaction assays are low. This suggests that the identified RNA-RNA interaction sites may be sequestered by local secondary structures. To investigate this further, we performed FCCS with ATTO647-labeled S10 RNA and a Cy3-labeled ssRNA probe, representing region 84–100 of S11 RNA (complementary to the region 272–288 of S10 RNA that corresponds to the peak shown in a blue box in *Figure 3*). No significant cross-correlation between S10 and a complementary RNA sequence was observed after incubation with NSP2 (*Figure 3—figure supplement 3*). In contrast, heat-annealing of a mixture of the two RNAs resulted in a high CCF amplitude (*Figure 3—figure supplement 3B and C*). This result strongly suggests that the identified sequence within S10 RNA is sequestered, and the free energy change of opening the intramolecular base pairs by NSP2 binding is too high. This idea is further corroborated by the analysis of secondary structure of S10 RNA (*Figure 3—figure supplement 4G*). Similar analyses of S1 and S4 sequences (*Figure 3—figure supplement 4*) suggest the identified S11 RNA-binding sites may be obstructed by local secondary structures that prevent interactions between S11 and those RNAs.

As a further control, we investigated the effects of mutations disrupting inter-molecular base-pairing between S11 and the interacting RNAs. We examined interactions between S11 and S5 by

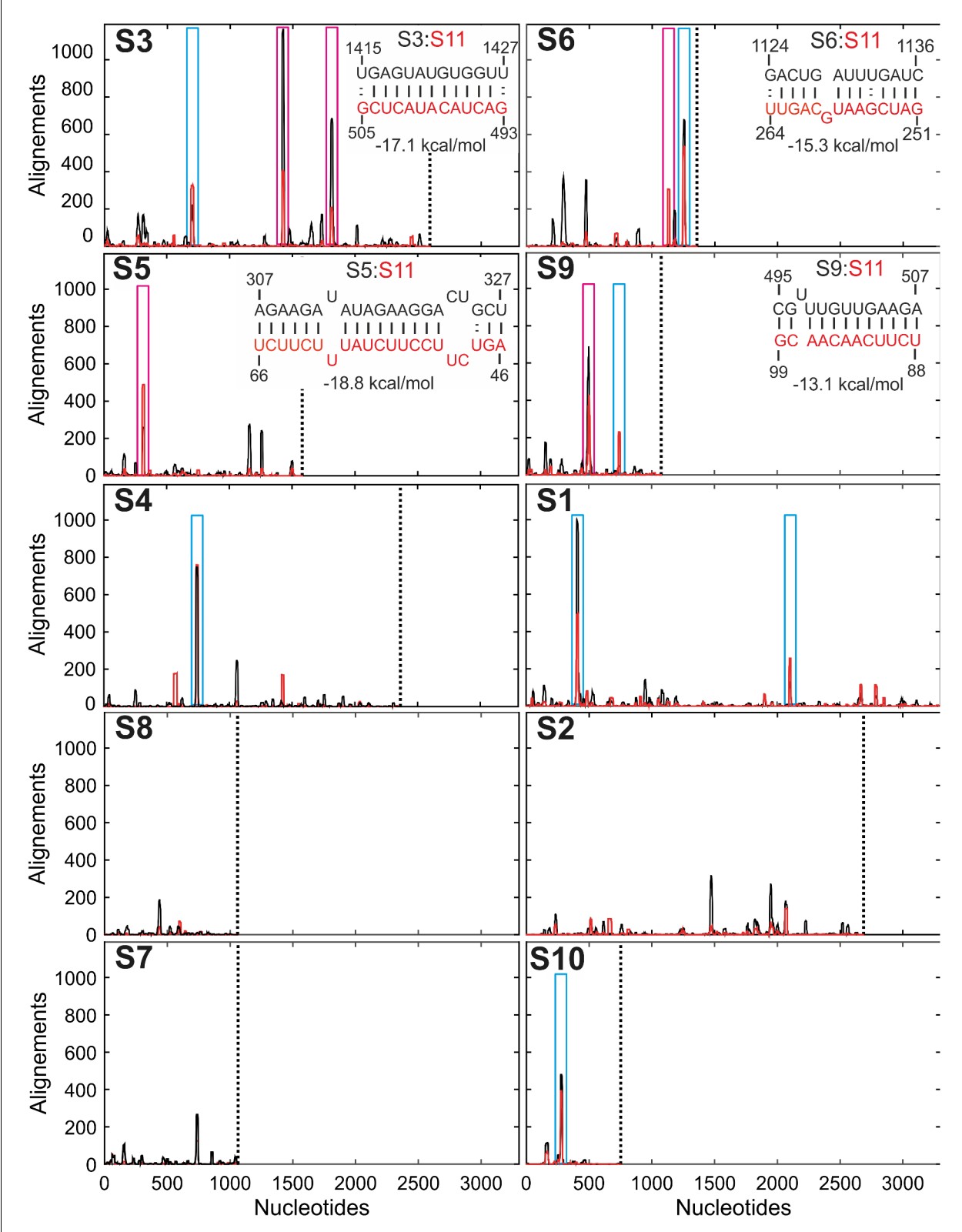

**Figure 3.** Inter-segment interactions between ssRNAs S11 and S1-S10, identified via RNA-RNA SELEX. SELEX-enriched sequences were aligned against S1-S10 genomic segments using probabilistic score-based filtering (Materials and methods) to construct the resulting histogram plots. Each histogram peak corresponds to a sequence with a potential to interact with the S11 RNA alone (shown as black peaks), or in the presence of NSP2 (peaks shown in red). These histograms are plotted in the order of decreasing CCF amplitudes (see *Figure 1B*). Each plot is scaled to the size of the longest segment

*Figure 3 continued on next page*

*Figure 3 continued*

S1 (3302 nts), and the 3'-end of each RNA segment is marked with a vertical dotted line. Sequences that can stably base-pair with S11 RNA in the presence of NSP2 are highlighted (magenta boxes). A number of genomic sequences identified in the RNA-RNA SELEX experiments may be sequestered by stable local secondary structures (peaks identified by cyan boxes). *Insets* – RNA-RNA interaction sites between RNA S11 (bottom strand shown in red), and its interacting partners S3, S6, S5 and S9 (top strand, shown in black).

DOI: https://doi.org/10.7554/eLife.27453.019

The following source data and figure supplements are available for figure 3:

**Source data 1.** Histogram plots (S1–S10) of SELEX-enriched sequences aligned against S1-S10 RNA segments using probabilistic score-based filtering (Bernoulli score of 14) from the RNA-RNA SELEX experiment against S11 RNA target.

DOI: https://doi.org/10.7554/eLife.27453.024

**Source data 2.** Histogram plots (S1–S10) of SELEX-enriched sequences aligned against S1-S10 RNA segments using probabilistic score-based filtering (Bernoulli score of 14) from the RNA-RNA SELEX experiment against S11 RNA in the presence of NSP2.

DOI: https://doi.org/10.7554/eLife.27453.025

**Figure supplement 1.** Sequence alignments of SELEX-enriched RNAs against a non-rotavirus control RNA (Non-RV).

DOI: https://doi.org/10.7554/eLife.27453.020

**Figure supplement 2.** RNA sequences identified in the RNA-RNA SELEX experiments, with a potential to interact with S11 RNA.

DOI: https://doi.org/10.7554/eLife.27453.021

**Figure supplement 3.** Accessibility of the region 272–287 of S10 RNA probed using a complementary RNA sequence, derived from S11 RNA.

DOI: https://doi.org/10.7554/eLife.27453.022

**Figure supplement 4.** The minimum free energy (MFE) secondary structure calculations of the complementary sequences with a potential for stable base-pairing with S11 RNA.

DOI: https://doi.org/10.7554/eLife.27453.023

introducing nucleotide substitutions within the identified RNA-RNA interaction sites (nts 307–322 in S5 and 51–66 in S11, *Figure 4A*).

Substitution of either RNAs with its mutated counterpart (S11mut or S5mut) resulted in a dramatic reduction of the CCF amplitude (*Figure 4A*, red and blue dotted lines), while the wild-type S5 and S11 RNAs formed stable complexes in the presence of NSP2 (*Figure 4A*, solid green line). Similarly, incubation of S11 with a non-RV control RNA resulted in low CCF amplitude (*Figure 4A*, dotted magenta line), reflecting its low propensity to form stable base-pairs with S11 RNA (*Figure 3—figure supplement 1*). Both combinations of S11 with S5mut and non-RV RNAs have similarly low CCF amplitudes, suggesting that the interacting RNAs did not stably interact. To further test this, we computer-generated a scrambled S11 RNA sequence with low propensity to form extensive base-pairing with S5 RNA (Materials and methods). Using FCCS, we probed the interactions between the 'scrambled' RNA and S5 RNA. The resulting apparent CCF amplitude was lower compared to the S5:S11mut combination (*Figure 4A*), suggesting that the observed weaker interactions are mediated by non-specific residual base-pairing between the mutated RNAs.

We then used the two-color FCCS approach to probe the stability of S5:S11 RNA complex. The FCCS analysis of the preformed S5:S11 RNA complex, incubated at elevated temperatures, yielded the estimated melting temperature, $T_m \sim 47°C$ (*Figure 4B*). This value is in good agreement with the $T_m$ of the identified RNA-RNA duplex, formed between S5 and S11 (*Figure 3*, inset, $T_m = 46°C$, calculated as described in Materials and methods).

We also introduced compensatory nucleotide substitutions in the putative interacting sequences of S11 and S5mut in order to test if the disrupted inter-segment contacts could be rescued by restoring the stable base-pairing between the mutated RNAs. The resulting mutated trans-complementary RNAs S5mut and comp_mutS11 strongly interacted with each other upon incubation with NSP2 (*Figure 4—figure supplement 1A and B*). Moreover, the CCF amplitude between the trans-complementary mutant RNAs was higher than that of the interacting wild-type RNA sequences, consistent with the increased stability of the rescued RNA-RNA contacts (*Figure 4—figure supplements 1B,* −18.8 kcal/mol for the S5:S11 complex and −23.7 kcal/mol for S5mut:S11comp). Collectively, these data demonstrate the importance of the identified sequences in the formation of stable and sequence-specific RNA-RNA contacts between S5 and S11.

We also carried out similar mutagenesis analyses for segment precursors S3 and S6 (*Figure 4—figure supplement 2*), confirming the identified RNA-RNA interaction sites in S11 RNA. All identified sequences (*Figure 3*, peaks highlighted in magenta) were capable of forming stable duplexes with

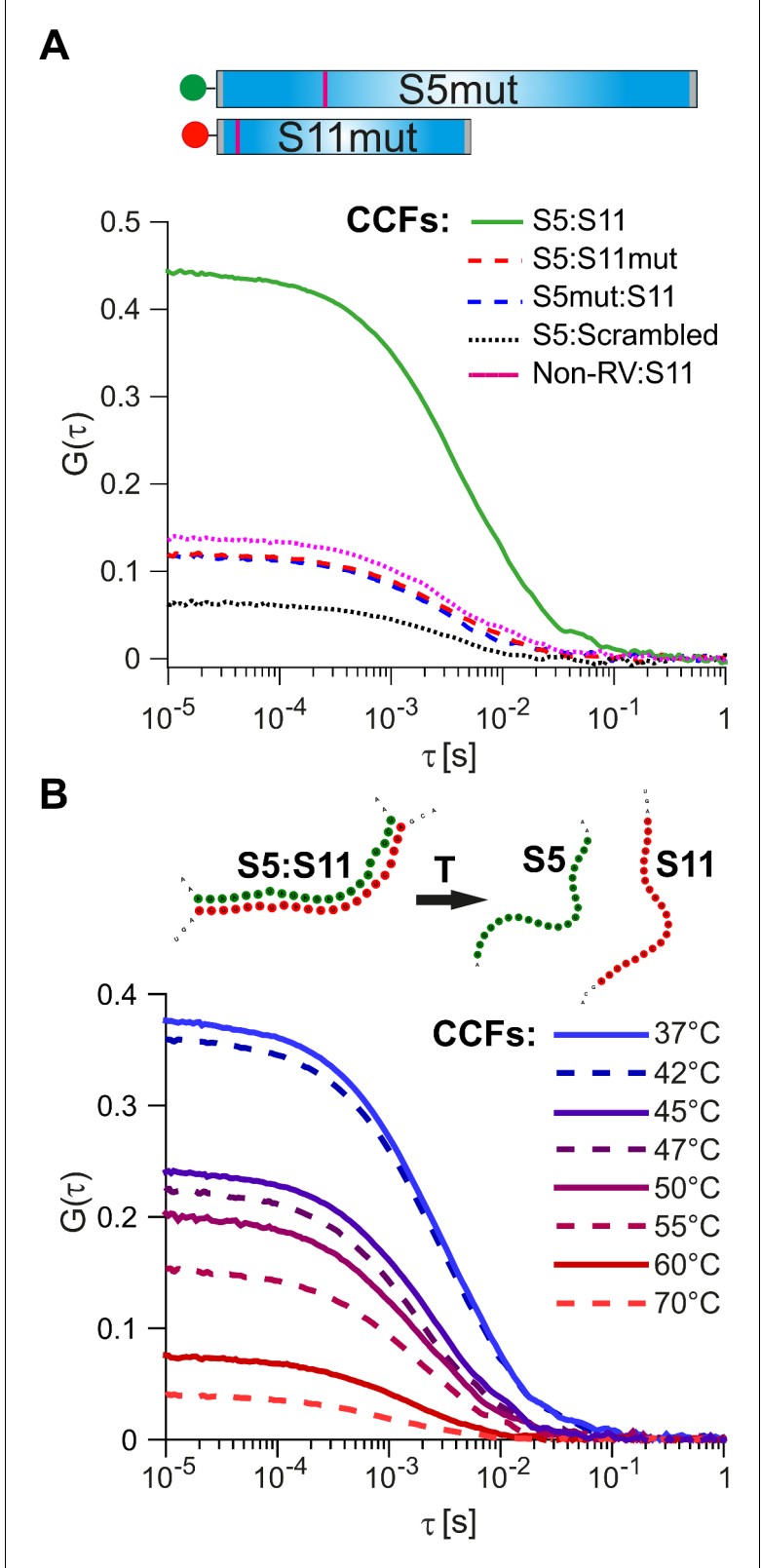

**Figure 4.** Probing the stability and specificity of inter-segment RNA-RNA interactions. (**A**) A schematic of the mutated RNAs S5 and S11 (S5mut and S11mut, respectively), showing the sites of mutations that disrupt inter-segment interactions (magenta stripes). These RNAs were examined in pairwise RNA-RNA interaction assays, as described in Materials and methods. Below – normalized CCFs of the wild-type S5:S11 RNA:RNA complexes (solid

*Figure 4 continued on next page*

*Figure 4 continued*

green line, high amplitude CCF), and significantly weaker interacting S11 and S5mut (dashed blue line), or S5 and S11mut ssRNAs (dashed red line). The non-rotavirus control RNA (Non-RV, see Materials and methods) does not strongly bind S11 RNA upon incubation with NSP2, resulting in low CCF amplitude (dotted magenta line). Similarly, incubation of a 'scrambled' RNA, designed to have a minimal propensity to base-pair with S5 RNA, yields the lowest CCF amplitude (black dotted line). (B) Temperature melting experiments of the pre-formed S5:S11 RNA-RNA complexes, examined by FCCS. The observed melting temperature ($T_m$) of 46°C agrees well with the estimated $T_m$ of the identified RNA duplex, formed between S5 and S11 RNAs upon incubation with NSP2 (*Figure 3*, inset).

DOI: https://doi.org/10.7554/eLife.27453.026

The following source data and figure supplements are available for figure 4:

**Source data 1.** Normalized CCFs of the wild-type S5:S11 RNA:RNA complexes, and weakly interacting S11 and S5mut, S5 and S11mut ssRNAs, S11:Non-RV RNAs, and S5 RNA:'scrambled' RNA.
DOI: https://doi.org/10.7554/eLife.27453.030
**Source data 2.** Normalized CCFs of the wild-type S5:S11 RNA:RNA complexes, pre-incubated at different temperatures to determine the Tm of the preformed RNA-RNA complex S5:S11.
DOI: https://doi.org/10.7554/eLife.27453.031
**Figure supplement 1.** Compensatory mutations rescue stable interactions between mutated ssRNAs S5 and S11.
DOI: https://doi.org/10.7554/eLife.27453.027
**Figure supplement 2.** Investigating RNA-RNA interaction sites between S3 and S11, and S6 and S11.
DOI: https://doi.org/10.7554/eLife.27453.028
**Figure supplement 3.** Putative sites of inter-segment interactions between S11 and S3, S5, S6 and S9.
DOI: https://doi.org/10.7554/eLife.27453.029

---

S11 RNA (insets in *Figure 3* and *Figure 3—figure supplement 2*), consistent with the high CCF amplitudes observed for the S3:S11 and S6:S11 complexes. Collectively, our data provide extensive experimental evidence that sequence-specific inter-molecular base-pairing mediated by NSP2 binding to viral ssRNAs, governs inter-segment interactions in RVs.

## Discussion

Several models of segmented genome encapsidation have been proposed to explain selective packaging of eleven RNAs in RVs (*Patton and Spencer, 2000*; *Trask et al., 2012*; *McDonald et al., 2016*). Here, we provide direct evidence of specific, inter-segment RNA-RNA interactions in RVs, by establishing experimental assays based on a combination of two-colour FCCS and an RNA-RNA SELEX. The FCCS-based assays allow rapid quantitative analyses of the stability of RNA-RNA and ribonucleoprotein complexes, providing additional information about the stoichiometry of the RNAs within such complexes. To delineate the sequences that mediate contacts between the RV ssRNAs, we have employed the RNA-RNA SELEX approach for identifying stable, specific inter-segment RNA-RNA interactions.

The data presented here collectively support the model in which RV ssRNAs can specifically interact with each other as a result of RNA remodeling brought about by the ssRNA-binding protein NSP2 (*Figure 5*). The results of RNA-RNA SELEX against S11 reveal multiple areas of the RNA undergoing conformational rearrangements upon NSP2 binding, while the most stable S11 intramolecular helices H1-H3 remain largely inaccessible. This model is consistent with the high affinity of NSP2 for ssRNA, but not dsRNA, which may underlie the protein's ability to alter the kinetics of RNA-RNA hybridization by accelerating the breakdown of weaker intra-molecular helices, concomitant with the stabilization of the alternative, stable inter-molecular contacts. Substitution of NSP2 with its mutant ΔC-NSP2 with significantly lower affinity for ssRNA reduces the efficiency of formation of inter-segment contacts, further supporting the proposed model of NSP2-mediated remodeling of RV (+)ssRNAs. Such a mechanism would account for NSP2-facilitated selection of thermodynamically favorable inter-segment interactions that may not always follow strict pairing rules. Because of the high affinity of NSP2 for ssRNA it is expected that NSP2 may remain associated with the oligomeric ssRNA complex prior its replication. Given that NSP2 inhibits the initiation of replication, but not the elongation stage (*Vende et al., 2003*), it is possible that the inner core protein VP2 binding to the ssRNA-NSP2 complex (*Guglielmi et al., 2010*; *Estrozi et al., 2013*;

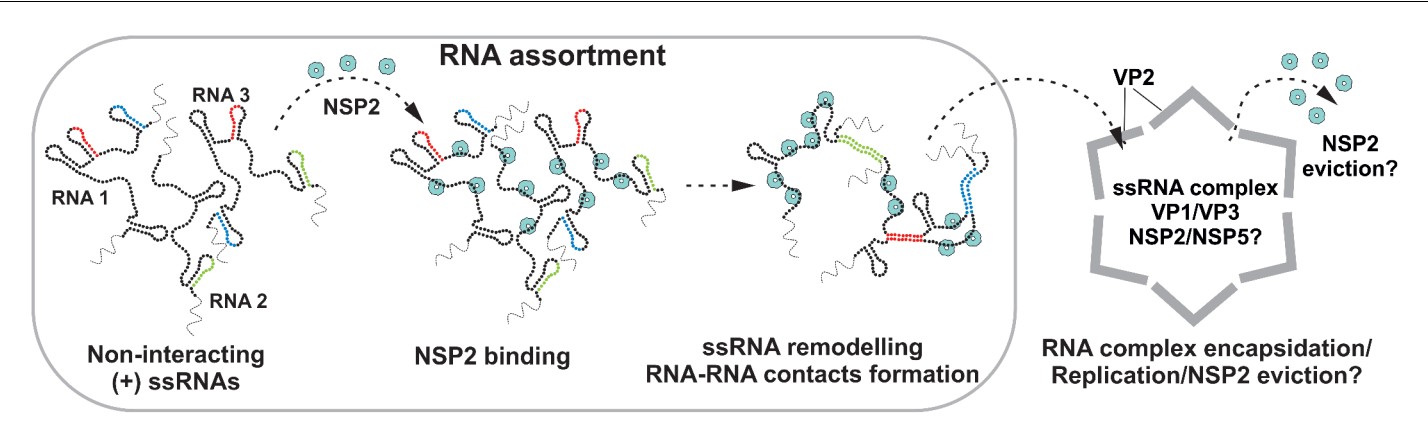

**Figure 5.** Proposed Model of the Mechanism of NSP2-mediated RNA Segment Assortment in Rotaviruses. RV segment (+)ssRNAs (here only 3 RNAs are shown schematically as RNA1, RNA2 and RNA3) do not form stable RNA-RNA contacts with each other, consistent with the previously reported lack of oligomerization of ssRNA transcripts emerging from transcribing RV double-layered particles (*Periz et al., 2013*). NSP2 (doughnut-like cyan octamers) binding to ssRNAs results in their structural rearrangements concomitant with the exposure of otherwise sequestered complementary sequences (interspersed sequences shown in red, blue and green) capable of inter-segment base-pairing. The exposed complementary sequences form stable sequence-specific inter-segment contacts (RNA helices shown in red, green and blue) during the RNA assortment process. The resulting multi-RNA ribonucleoprotein complex (containing NSP2/NSP5 and multiple copies of the viral polymerase VP1/capping enzyme VP3, not shown here) may nucleate the VP2 core assembly around it. In this model, the core assembly is concurrent with further structural reorganisation of the RNA assortment complex, displacement of ssRNA-bound NSP2 octamers and (-)strand RNA synthesis by VP1.
DOI: https://doi.org/10.7554/eLife.27453.032

*Patton, 1996*) would result in the initial displacement of the NSP2 during the core assembly nucleation, followed by further eviction of NSP2 during the elongation. In this model, the assembly of the core is nucleated by multiple copies of the RV polymerase VP1 bound to eleven pre-selected (+) ssRNAs (*Patton, 1996*), arranged within the ssRNA-NSP2 complex at defined spatial locations. The highly basic N-terminal 'arms' of VP2 would interact with VP1 and ssRNA (*Zeng et al., 1998*; *Guglielmi et al., 2010*; *Estrozi et al., 2013*) within the ribonucleoprotein complex, which could also facilitate the formation of the VP2 lattice around it. The proposed model would account for incorporation of multiple assorted (+)ssRNAs, each associated with its own copy of the VP1 polymerase (*Guglielmi et al., 2010*; *Patton, 1996*; *Periz et al., 2013*). Interestingly, at high concentrations VP2 self-assembles into empty core-like particles (*Crawford et al., 1994*). This implies that VP2 nucleation must be tightly regulated to prevent early assembly to ensure specific packaging of eleven (+) ssRNAs (*Desselberger et al., 2013*; *Viskovska et al., 2014*). It is possible that nucleation of core assembly by (+)ssRNA complexes may occur at VP2 concentrations significantly lower, than those required for formation of empty cores (*Borodavka et al., 2012*). Binding of NSP2 may also contribute to the regulation of over-nucleation of VP2 on the RNA substrate (*Viskovska et al., 2014*). Early biochemical studies of RV assembly intermediates suggest that (+)ssRNA replication may take place within the pre-core intermediates lacking a complete VP2 shell (*Gallegos and Patton, 1989*). The proposed model would explain the absence of NSP2 inside the assembled virions, despite its abundance in viroplasms and high affinity for ssRNAs (*Figure 5*).

Our studies also reveal several parallels in the ways, by which segmented RNA viruses, including RVs, Bluetongue virus (BTV) and influenza A viruses (IAVs), may control selection of the correct RNA segments (*Fajardo et al., 2015*; *Gerber et al., 2014*). In BTV, multiple sites of the smallest segment S10 RNA have been shown to be involved in assortment of other genomic segments (*Fajardo et al., 2015*). In IAVs, genome segment packaging into virions is coordinated via interactions among the different viral RNA segments, while the sites of inter-segment interactions overlap with the coding regions of segmental RNAs (*Gavazzi et al., 2013b*; *Gerber et al., 2014*). Our data suggest that the majority of RNA-RNA interaction sites are also located within the coding regions of the RV genomic segments (*Figure 4—figure supplement 3*). The IAV nucleoprotein (NP) has also been shown to regulate selective packaging of the viral genomic segments (*Moreira et al., 2016*), revealing the distinct amino acid residues of NP that are required for packaging of specific RNA segments. Interestingly,

the RNA-RNA interaction assays conducted in the presence of group C rotavirus NSP2 did not yield stable inter-segment RNA contacts normally formed upon co-incubating interacting ssRNAs with group A rotavirus NSP2 (*Figure 1—figure supplement 5B*). Previously, using a NSP2 complementation system Taraporewala *et al.* reported that group C NSP2 could not rescue replication in NSP2-deficient cells infected with group A RV (*Taraporewala et al., 2006*). Our assays present additional evidence that despite multiple structural similarities and analogous enzymatic activities, group C RV NSP2 not only fails to substitute group A RV NSP2 during assembly of viroplasms, but is also incompetent in promoting the formation of stable RNA-RNA contacts between segmental ssRNAs.

Interestingly, the IAV segment ssRNAs were shown to form higher order species in vitro; however most of these interactions were only detectable upon co-incubation of the RNAs at 55°C (*Gavazzi et al., 2013a*). Similarly, BTV ssRNAs would also interact with each other, albeit with extremely low efficiencies, unless they were co-transcribed in vitro (*Fajardo et al., 2015*). Using the FCCS-based approach, we did not detect inter-segment interactions prior the incubation with NSP2, consistent with previously reported lack of association of the RNAs, emerging from the transcribing rotavirions (*Periz et al., 2013*).

Although the precise mechanism of the selective packaging of eleven distinct RNAs in RVs is still largely unknown, our results pave the way for delineating the mechanistic details and the RNA sequences essential for this process. Combining our approach with a recently established reverse genetics system for RVs (*Kanai et al., 2017*) will ultimately move us one step closer to understanding segment selection and encapsidation in these important pathogens. However, currently the analysis of such interactions in vivo is unfeasible, mostly due to the low rescue efficiency of the rotavirus reverse genetics system, through which only a very limited number of mutated genomic sequences have been rescued at this point (T.Kobayashi, personal communication). In conclusion, the experimental approach reported here allows identification of RNA-RNA interactions without the need for cross-linking and intercalating agents, which may perturb dynamic RNA structures. It takes into account the kinetic effects of structure formation, e.g., when RNA binding is concomitant with structural changes due to binding of other RNAs, or proteins. The FCCS-based assays presented here provide instant quantitative validation of RNA-RNA interactions and probe their stabilities. This approach can be used for high-throughput screening of the conditions that favor stabilization of RNA-RNA contacts, thus providing an additional dimension to the panoply of techniques for detecting base-paired RNA nucleotides. The combination of FCCS and RNA-RNA SELEX-based approaches holds promise for validation and quantitative analyses of the interacting viral ssRNAs and miRNAs. We anticipate that these methods will become useful tools in the armory of instruments for gaining novel mechanistic insights into the dynamics of genomic RNA-RNA interactions in RNA viruses.

## Materials and methods

### Plasmids

Genome segment precursors S1-S11 (GenBank IDs are listed in *Supplementary file 2*) of bovine rotavirus strain RF (G6P6[1]) were obtained as pUC19 cDNA clones (*Richards et al., 2013*) from Dr Ulrich Desselberger and Professor Andrew Lever (University of Cambridge, UK). Mutant RNA sequences were produced from the original pUC19 plasmids harboring S11, S5 and S6 cDNA clones using the Q5 site-directed mutagenesis kit (NEB) and the oligonucleotides listed in *Supplementary file 2*. A DNA sequence was designed to make an RNA template with a minimal propensity to stably form base pairs (longer than 8 bp) with S5 and S6 RNAs. S11 RNA was used as an input sequence. It was mutated into a sequence with a minimal number of substitutions required to obtain a 'scrambled' sequence, which has only short (maximum of 8 nt) regions of complementarity with the sequences of S5 and S6 RNAs. The 'scrambled' 667-nt long RNA sequence, lacking extended regions of complementarity with S5 and S6 (*Supplementary file 2*), was cloned into a pUC19 vector, as previously described (*Borodavka et al., 2016*). The non-rotavirus control RNA (Non-RV) was transcribed as a 1221-nt long Satellite Tobacco Necrosis Virus (STNV) genomic RNA from the DNA template as previously described (*Borodavka et al., 2012*).

## NSP2 and NSP5 expression and purification

cDNA clone of gene 8 of rotavirus A (strain RF) was used to PCR-amplify NSP2 ORF (*Supplementary file 2*) with NcoI and XhoI restriction sites used for ligating the resulting double-digested NSP2-coding fragment into a linearized pET-28b vector. The resulting pET-28b-NSP2 construct was verified by sequencing and used for protein expression in BL21(DE3) *E.coli* as previously described (*Schuck et al., 2001*). NTA-affinity purified NSP2 fractions were further purified over a HiTrap SP cation-exchange column (*Schuck et al., 2001*). The concentrated peak fractions were resolved on a Superdex 200 10 × 300 GL column and pre-equilibrated with RNAse-free SEC buffer (25 mM HEPES-Na, pH 7.5, 150 mM NaCl) to ensure high purity and homogeneity of the preparation. Rotavirus group C NSP2 (strain Bristol) was expressed from pQE60g8C construct and purified, as described in *Taraporewala et al. (2006)*. Bovine rotavirus A (strain RF) protein NSP5 was expressed and purified as previously described in (*Martin et al., 2011*). A plasmid for expression of the C-terminally truncated NSP2 variant ΔC-NSP2 was constructed using pET-28b-NSP2 vector by removing C-terminal residues 295–317 (*Hu et al., 2012*). The resulting ΔC-NSP2 variant was expressed and purified following the purification procedures for a full-length NSP2, as described above.

## RNA transcription and labeling

RNA transcripts were produced by in vitro transcription of linearized DNA templates (*Supplementary file 2*) using the HiScribe T7 RNA synthesis kit (NEB). Transcribed RNAs were purified using RNeasy columns (QIAGEN). All in vitro transcription reactions were carried out using the T7 RNA transcription kit (NEB), following the manufacturer's protocol, except for the fluorescently labeled or biotinylated RNA samples. RNA transcripts were labeled by introducing 5′-aminoallyl-uridine-5′-triphosphate (5-AA-UTP, Thermo Fisher) during the in vitro transcription. Similar uridine derivatives with substituents at the fifth carbon have been previously shown to support the replication of RVs (*Silvestri et al., 2004*). This minimally perturbing RNA labeling approach results in only few 5-AA-uridines incorporated per each transcript. Such a labeling strategy minimizes the influence of fluorescence quenching or Förster Resonance Energy Transfer (FRET) between the reporter dyes on the apparent cross-correlation amplitudes (*Foo et al., 2012*). For labeling, amine-modified RNAs were produced by incorporating 5′-aminoallyl-uridine-5′-triphosphate (5-AA-UTP) (3:1 ratio UTP:5-AA-UTP) during T7 transcription (*Borodavka et al., 2012*). Amine-modified RNAs (1–2 µM), in a total volume of 100 µL, RNAse-free 100 mM sodium borate buffer (pH 8), were reacted with 1 mM ATTO647- or ATTO565-NHS ester for 2 hr at 4°C. Under physiological conditions, the chosen dyes are characterized by zero net charge, relatively high hydrophilicity and high photo- and thermal stabilities, important for RNA melting experiments. Fluorescently labeled and biotinylated RNAs were purified using RNeasy columns, as previously described (*Borodavka et al., 2012*). RNA labeling efficiencies were routinely examined spectrophotometrically (*Borodavka et al., 2012*). Based on the estimated labeling efficiencies, typically 85% of all RNA molecules contained 1–6 dye molecules and less than 2% of RNAs were unlabeled. RNA biotinylation was carried out under similar conditions except EZLink Sulfo-NHS-LC-LC-biotin was used in lieu of NHS-esters of ATTO dyes. Further purification of biotinylated RNAs was carried out following the purification protocol used for dye-labeled RNAs (see above). All RNA samples were routinely examined on denaturing formaldehyde agarose gels to ensure their integrity (*Figure 2—figure supplement 2B and C*).

## Fluorescence cross-correlation spectroscopy (FCCS)-based assays of RNA-RNA interactions

For FCCS, to minimize any potential RNA self-association (*Figure 2—figure supplement 2D*), all RNA samples were heat-annealed for 5 min at 70°C in 10 mM HEPES-Na, pH 7.4, slowly cooled and diluted in an assay buffer (100 nM – 1 µM RNA in 20 mM HEPES-Na, pH 7.4, 1 mM MgCl$_2$, 150 mM NaCl, 0.05% Tween 20, 1 mM DTT).

Reactions were set up with equimolar amounts of ATTO647-dye labeled S11 RNA or 'scrambled' or mutated S11 RNA sequence and ATTO565-dye labeled RNA Sx (where x is any other interacting RNA partner). RNA samples were mixed in reaction buffer (0.1–2.5 µM total concentration) and allowed to interact for 30 min at 37°C. For NSP2-mediated RNA-RNA-interaction assays, labeled RNA samples (150 nM total RNA concentration) were incubated with 2.5–10 µM NSP2 (or 2.5–10 µM

BSA for control reactions). Reactions were allowed to proceed at 37°C for 30 min before they were stopped by diluting RNA samples into assay buffer, as discussed above, to achieve a final concentration of labeled RNAs of 1 nM (each strand). These samples were further allowed to equilibrate at 25°C for 15 min prior to FCCS measurements. Proteinase K digestion of NSP2 was performed by adding 40 µg of proteinase K (800 U/ml) to the NSP2-RNA samples and incubating them for 30 min at 37°C, as previously described (*Taraporewala and Patton, 2001*), prior to dilution into assay buffer as described above. Temperature melting experiments were performed by measuring the FCCS amplitudes of the pre-formed S5:S11 RNA-RNA complex (1 nM each RNA strand), treated with proteinase K. S5:S11 RNA-RNA complex was diluted into an assay buffer without MgCl$_2$. The diluted RNA samples (1 nM of each strand) were incubated at 37-70°C for 15 min, after which they were slowly cooled down to 25°C prior to FCCS measurements. S10 RNA oligonucleotide hybridization assay was performed by heat-annealing RNA substrates (200 nM of ATTO647-labeled S10 RNA and Cy3-labelled RNA oligonucleotide Seq11_84_100, as described in *Supplementary file 2*) for 5 min at 70°C in 100 mM NaCl, 20 mM HEPES-Na buffer, pH 7.4, and slowly cooling down and diluting the complexes in assay buffer as described above.

## Dual-color fluorescence cross-correlation spectroscopy (FCCS) data acquisition and analysis

Dual-color fluorescence cross-correlation spectroscopy is a technique that allows sensitive detection of interactions of differently labeled molecules at low concentrations, typically with single-molecule sensitivity (*Figure 1—figure supplement 1*). FCCS detects coincident fluctuations in both the green and red channels. The amplitude of the cross-correlation function (CCF) is proportional to the number of double-labeled complexes and inversely proportional to the average number of labeled species in each individual channel (i.e., red and green). Therefore, the fraction of double-labeled species in a sample can be extracted from the ratio of the amplitudes of the cross-correlation (CCF) to the auto-correlation functions (ACF). Conventionally, in dual-color FCCS, the signal from differently labeled molecules is divided into channels by emission (e.g., via dichroic mirrors and filters) (*Schwille et al., 1997*; *Bacia and Schwille, 2007*). Due to spectral crosstalk, an artificial cross-correlation signal can be observed between the two channels, even when the different species do not interact. When using pulsed interleaved excitation (PIE) in combination with the high temporal resolution of time-correlated single photon counting (TCSPC), the signals can be further divided by the excitation source. This removes the influence of the crosstalk and greatly enhances both the sensitivity and specificity of this technique (*Müller et al., 2005*).

FCCS measurements of ATTO647, and ATTO565 or Cy3-dye labeled RNAs were performed on a custom-built confocal microscope with time-correlated single photon counting (TCSPC) detection designed for pulsed interleaved excitation (PIE), as previously described (*Müller et al., 2005*). Two pulsed lasers at 561 nm (frequency-doubled fiber laser, Toptica Photonics) and 635 nm (diode laser LDH-P-C-635b, Picoquant) with a fixed repetition rate of 27.4 MHz were used for excitation with the laser power set to 10 µW, as measured before the objective. The emission of the red laser was delayed electronically by 20 ns with respect to the yellow laser to achieve pulsed interleaved excitation. A 60x, 1.27 NA objective (Plan Apo IR 60 × WI, Nikon) was used both to focus the excitation light and to collect the fluorescence. The fluorescence was focused on an 80 µm pinhole to remove out-of-focus light. Green and red emission signals were spectrally separated and focused on two single photon avalanche photodiodes (APDs). The photon detection signals of each APDs are timed and recorded with separate TCSPC cards (SPC-150, Becker and Hickl).

Data analysis and processing were performed using a custom-written Matlab-based data processing platform PAM (PIE Analysis with Matlab, https://gitlab.com/PAM-PIE/PAM; a copy archived at https://github.com/elifesciences-publications/PAM-PIE-PAM), which includes tools for processing single point data collected using PIE (*Schrimpf et al., 2017*).

The ACFs and CCFs were fit with a one-component model showing dark state dynamics (*Bacia and Schwille, 2007*):

$$G(\tau) = \frac{\gamma}{N} \cdot \left( 1 + \frac{T}{1-T} \cdot e^{-\frac{\tau}{\tau_T}} \right) \cdot \frac{1}{1 + \frac{4D\tau}{\omega_r^2}} \sqrt{\frac{1}{1 + \frac{4D\tau}{\omega_z^2}}} + G(\infty)$$

Here, $N$ is the number of individually diffusing molecules in the focus volume, $D$ is the diffusion

coefficient, and $\omega_r$ and $\omega_z$ are the distances from the center of the focus to the point where the fluorescence intensity has decayed to $1/e^2$ of the maximum intensity for the lateral and axial dimensions, respectively. $T$ and $t_T$ denote the dark state fraction and the correlation time, respectively. $G(\infty)$ is a constant offset accounting for slow signal fluctuations. The geometric factor, $\gamma$, accounts for gradual intensity decrease of the assumed 3D Gaussian shape of the observation volume and is 0.35355. The CCF amplitudes were normalized by the $N$, measured for the ATTO647 labeled S11 RNA (or its mutated variants, including the 'Scrambled' RNA sequence). These normalized CCFs yield the fraction of the interacting ATTO565- and ATTO647-dye labeled ssRNAs. Thus, the fraction of interacting species is only an estimate of the absolute CCF value, as the stochastic labeling of the RNAs leads to the non-trivial weighting of a different number of fluorophores. In the case of complete binding with 1:1 stoichiometry, the ratio of the CCF/ACF amplitudes is expected to be close to one (*Foo et al., 2012*). The lower absolute CCF amplitudes are attributed to small differences in the observation volumes for red and green fluorophores.

RNA samples were measured at least three times, and the photon data were acquired for 10–15 min (or up to 45 min when the stability of the preformed RNA-RNA complexes were investigated). The resulting ACF and CCF amplitudes were averaged, yielding mean CCF amplitudes reported here.

## RNA-RNA SELEX (systematic evolution of ligands by EXponential enrichment)

A naïve 30-mer RNA library (N30) consisting of ~$3.6 \times 10^{13}$ N30 random sequences flanked by primer sequences for RT-PCR amplification and transcription was produced by in vitro T7 transcription from a DNA template (*Supplementary file 2*). It was purified on Agencourt AMPure beads (Beckman) following the manufacturer's protocol. Biotinylated S11 RNA target was prepared as described above. RNA target (200 nM) was heat-annealed for 5 min at 70°C in a low ionic strength buffer (10 mM HEPES-Na, pH 7.4), and slowly cooled to room temperature to minimize its oligomerization (*Li et al., 2010*). A naïve N30 RNA library (10 μM), re-suspended in SB buffer (10 mM HEPES-Na, pH 7.2, 150 mM NaCl, 1 mM DTT, 1 U/μL murine RNAse inhibitor (NEB), 1 mM MgCl$_2$, 0.1 mg/ml bovine serum albumin, BSA) was incubated with the S11 RNA target for 15 min at 37°C, either in the absence (RNA alone selection) or presence of 10 μM NSP2 (NSP2 +selection). 20 μL of RNAse-free Dynabeads MyOne Streptavidin T1 beads (Invitrogen) were re-suspended in SB buffer and added to the S11 RNA samples followed by further incubation at 37°C for 15 min. Captured RNA targets with bound 30-mer RNAs were washed with SB buffer (200 μl) 4 times for 10 min at 37°C during the first 3 rounds of selections, and the stringency of selection was increased after round 3 by increasing the number of washes up to 8 per round. Negative selections were carried out at every second round of SELEX using streptavidin-coated beads. S11-bound 30-mers were recovered by heat-elution (95°C, 5 min) of the RNAs from carrier beads into 30 μl of RNAse-free water. Eluted RNA libraries were reverse-transcribed using Superscript III Reverse Transcriptase (Thermo Scientific) and primer P1, and further PCR-amplified using primers P1 and P2 (*Supplementary file 2*). PCR-amplified SELEX-enriched cDNA libraries were analyzed by native PAGE after each group of two rounds of selection to confirm the presence of selected products for the next round of in vitro transcription and selection. In the final selection round, the resulting PCR products were prepared for Illumina MiSeq sequencing (LIMM DNA Sequencing Facility, University of Leeds).

## Identification of sites of RNA-RNA interactions

We identified 4,857,793 unique sequences from the RNA alone SELEX data and 3,308,107 sequences from the NSP2+-enriched data. The lower number of the unique sequences from the NSP2 +selection is due to selective enrichment of certain RNA sequences in the presence of NSP2. Each sequence from both data sets was aligned against all 10 segments (S1-10) and the single best alignment was identified using a probabilistic score (the Bernoulli score), which benchmarks the probability of a non-contiguous alignment to that of a contiguous alignment of N nucleotides (*Shakeel et al., 2017*). Thus, a 30-nt long RNA sequence, which aligns to the genomic segment with a total of N non-contiguous matches (N ≤ 30), will have a Bernoulli score of N. The score N is equivalent to the probability that a short sequence of N nucleotides would match the sequence of the genomic segment (*Shakeel et al., 2017*). We used the alignments with the Bernoulli score of 12 or

above to analyze sequences that strongly interact with S11 RNA to identify highly accessible areas in S11 RNA. Higher scores (14 or above) increase the stringency of the analysis (see *Figure 3—figure supplement 1*). We used RNA sequences with Bernoulli scores of 12 and above (for alignments against S11 RNA target) or 14 and above (for other RNA segments) to construct histogram plots for each genomic segment S1-S10. The histogram bins are associated with individual nucleotides in each segment. Each sequence aligning to a nucleotide increments the histogram bin by one, so that the peak height corresponds to the number of sequences aligning to it. This approach allows identification of multiple RNA sequences that strongly interact with S11 RNA, including RNAs not fully matching the S1-S10 genomic sequences to account for nucleotide mismatches and possible gapped regions. We used the areas of high sequence alignments to identify regions that have a high probability of being single-stranded. These data were used as constraints for computing secondary structures of S11 RNA, with and without NSP2, using Mfold (*Zuker, 2003*).

## Circular Dichroism (CD) Spectroscopy of S11 RNA

CD spectra were acquired between 240 nm and 320 nm in a 1 cm-long path cell, as previously described (26). CD spectra were recorded for the S11 RNA (200 nM in 10 mM HEPES-Na, pH 7.6) before and after incubation with either 10 µM NSP2 or RNAse-free acetylated bovine serum albumin (BSA, New England Biolabs) for 15 min. RNA secondary structure transitions upon thermal melting were also monitored at various temperatures up to 90°C.

## RNA structure analysis and visualization

RNA secondary structures were predicted using minimum free energy (MFE) modeling in RNAfold to compute the thermodynamic ensemble of secondary structures and base-pairing probabilities. The centroid structures showing minimal base-pair distances to all other secondary structures in the Boltzmann ensemble were analyzed in order to identify regions of high and low accessibilities within S1-S10 RNAs. Similarly, the MFE structure of S11 RNA was calculated in Mfold (RRID:SCR_008543) with the folding restraints derived from the RNA-RNA SELEX experiments. Differences in target RNA accessibilities were expressed as a normalized number of sequence alignments in the presence of NSP2, after subtracting alignments, identified in the S11 structure alone. These values were used to calculate a color map, applied to the S11 RNA structure, shown in *Figure 2C*.

   RNA structures were visualized in VARNA (*Darty et al., 2009*), and jVizRNA 2.0 was used for generating secondary structure circular plots (*Wiese et al., 2005*). The free energies of hybridization of the interacting sequences were calculated using IntaRNA (Freiburg RNA tools) with the ensemble free energy calculations realized in Vienna RNA library (*Lorenz et al., 2011*; *Wright et al., 2014*). Duplex melting temperature ($T_m$) calculations were performed using nearest-neighbor parameters implemented in MELTING v.5.0 (56), estimated for 1 nM RNA strand concentration using the parameters matching the low-salt buffer conditions, used for FCCS measurements, as described above.

## Acknowledgements

We are indebted to Dr Ulrich Desselberger, Dr James Richards and Prof. Andrew Lever (University of Cambridge) for the generous gift of the full-length cDNA copies of RV genomic segments for in vitro transcription; Dr John Patton for sharing a DNA clone pQE60g8C for expressing group C rotavirus NSP2. The authors would like to thank Dr Ulrich Desselberger (University of Cambridge) for critical reading of the manuscript; Prof. Takeshi Kobayashi (University of Osaka) for his valuable advice and discussions on the rotavirus reverse genetics system established in his laboratory. We thank Dr John T. Patton (University of Maryland), Prof Peter Stockley, Dr Roman Tuma and Mr Jack Bravo (University of Leeds) for valuable discussions and comments during the preparation of the manuscript.

## Additional information

### Funding

| Funder | Grant reference number | Author |
|---|---|---|
| Wellcome | 103068/Z/13/Z | Alexander Borodavka |
| Leverhulme Trust | ECF/019/2013 | Eric C Dykeman |
| Deutsche Forschungsge-meinschaft | SFB1032 | Waldemar Schrimpf<br>Don C Lamb |

The funders had no role in study design, data collection and interpretation, or the decision to submit the work for publication.

### Author contributions

Alexander Borodavka, Conceptualization, Resources, Data curation, Formal analysis, Supervision, Funding acquisition, Validation, Visualization, Methodology, Writing—original draft, Project administration, Writing—review and editing; Eric C Dykeman, Resources, Data curation, Software, Formal analysis, Methodology, Writing—review and editing; Waldemar Schrimpf, Data curation, Formal analysis, Visualization, Methodology, Writing—original draft, Writing—review and editing; Don C Lamb, Resources, Software, Funding acquisition, Writing—review and editing

### Author ORCIDs

Alexander Borodavka (iD) https://orcid.org/0000-0002-5729-2687

### Decision letter and Author response

Decision letter https://doi.org/10.7554/eLife.27453.040
Author response https://doi.org/10.7554/eLife.27453.041

## Additional files

### Supplementary files

• Supplementary file 1. Estimated diffusion coefficients (D) and hydrodynamic radii ($R_h$) of the fluorescently labelled RV segment precursors ssRNAs. RNA samples were measured at 25°C in 10 mM HEPES-Na, pH 7.4, 150 mM NaCl, 0.5 mM $MgCl_2$. For each RNA sample, photon data were acquired and 20 autocorrelation functions (ACFs) were calculated, as described in Materials and methods. The ACF data were fitted to a one-component diffusion model with dark state dynamics (Materials and methods) to calculate their respective diffusion time values for each RNA sample, and their corresponding diffusion coefficients and hydrodynamic radii were computed assuming the diffusion coefficient of the free ATTO488 dye (D = 400 $\mu m^2$/s at 25°C), as described in Materials and methods. Diffusion coefficients and hydrodynamic radii are reported as mean ±SD for at least 3 RNA samples preparations.
DOI: https://doi.org/10.7554/eLife.27453.033

• Supplementary file 2. DNA and RNA sequences used in the study. DNA oligonucleotides used for site-directed mutagenesis and cloning purposes and RNA oligonucleotides used for FCS measurements. Genbank accession numbers are given for genome segment cDNA templates used for the in vitro transcription of S1-S11 and Non-RV control RNAs. All RNA and DNA oligonucleotides and gBlock DNA sequences were obtained commercially from Integrated DNA Technologies.
DOI: https://doi.org/10.7554/eLife.27453.034

• Source code 1. Source code for PAM - PIE Analysis with MATLAB, a GUI-based software package used for single-molecule fluorescence data analysis.
DOI: https://doi.org/10.7554/eLife.27453.035

• Transparent reporting form
DOI: https://doi.org/10.7554/eLife.27453.036

## Major datasets

The following dataset was generated:

| Author(s) | Year | Dataset title | Dataset URL | Database, license, and accessibility information |
| --- | --- | --- | --- | --- |
| Borodavka A, Dykeman E, Schrimpf W, Lamb D | 2017 | Data from: Protein-mediated RNA folding governs sequence-specific interactions involved in genome segment selection in rotaviruses | http://dx.doi.org/10.5061/dryad.vp312 | Available at Dryad Digital Repository under a CC0 Public Domain Dedication |

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
