## [Decision Letter]

Thank you for submitting your article "Protein-mediated RNA folding governs sequence-specific interactions involved in genome segment selection in rotaviruses" for consideration by *eLife*. Your article has been reviewed by two peer reviewers, and the evaluation has been overseen by a Reviewing Editor and Wenhui Li as the Senior Editor. The following individual involved in review of your submission has agreed to reveal his identity: Bidadi Venkataram Prasad (Reviewer #2).

The reviewers have discussed the reviews with one another and the Reviewing Editor has drafted this decision to help you prepare a revised submission.

Summary:

This manuscript tries to address the fascinating and complex question of how eleven segments of RNA are selected for subsequent encapsidation in rotavirus. Rotavirus is a member of the Reoviridae. A common feature is that these viruses encapsidate 9–12 segments of dsRNA; rotaviruses, in particular, have 11 dsRNA segments. How these viruses in the Reoviridae selectively package a correct set of RNA segments has been an unanswered question. Borodavka et al., present an interesting set of experiments to address how the 11 segments of the rotavirus RNA genome are selected for packaging into virions. The data suggest that the rotavirus non-structural protein NSP2 remodels viral ssRNA to facilitate the formation of stable inter-segment contacts. In particular, authors show that segment S11 (smallest of the segments) interacts with S10 and this association is further stabilized by the presence of other RNA segments and that S11 preferentially binds to S3, S5, and S6. Authors further show that these specific inter-segmental interactions result in the formation of an RNA oligomer which remains stable even in the absence of NSP2, thus suggesting that the role of NSP2 is to promote such specific inter-segmental contacts and facilitate the formation of an RNA oligomer presumably containing the correct set of 11 segments for subsequent packaging. Using RNA SELEX methods authors have mapped the sequences in S11 that are likely involved in the inter-segmental RNA-RNA base-pairing, which were further substantiated selective mutating the contact regions. Overall this is an elegant study, well designed, presenting novel experimental approaches. Previous studies have suggested or speculated the possibility of such interactions as a requirement of selective packaging of RNA segments, however, this is perhaps the first study to demonstrate the possibility of such inter-segmental interactions in the case of rotavirus.

Overall, this is a very nice paper demonstrating the possibility of specific inter-segmental interactions among rotavirus RNAs that is promoted by NSP2 using a clever combination of various techniques. It is unfortunate that the reverse genetics system published recently for rotaviruses is not sufficiently robust to address the functional relevance of the identified interactions. However, even in the absence of such data, the current manuscript makes a significant step towards uncovering the molecular basis of RNA packaging in rotaviruses and the presented methodologies will be useful for analysing RNA packaging mechanisms in other segmented negative strand RNA viruses.

Essential revisions:

The data in Figure 4 would be considerably strengthened by showing that interactions between S5:S11mut can be recovered by compensatory mutations in S5 and, vice versa, that S5mut:S11 interactions can be recovered by compensatory mutations in S11. Similar interaction 'rescue' experiments could be carried out for the S3:S11 and S6:S11 pairs presented in Figure 4—figure supplement 1.

One major concern of this paper is the lack of in vivo validation for the observations made, and a lack of a well-reasoned model for how the formation of the higher-order ssRNA-oligomer allows itself for replication, duplex formation, and packaging. As noted above, we understand that in vivo validation is not practical at this point. Still, there are several questions (some of which could be addressed by in vitro experiments as well) such as for example 1) given that NSP2 strongly interacts with NSP5 in the viroplasm (and perhaps with other proteins including VP1 and VP2), how do these interactions affect NSP2-RNA interactions and RNA remodeling? Can the experiments be done in the presence of both NSP2 and NSP5? Do NTPase/RTPase activities of NSP2 play a role in RNA-RNA interactions or remodeling? This can be tested perhaps by using a mutant of NSP2 that lacks NTPase/RTPase activities? Addition of NSP5 and/or elucidation of potential roles of NTPase/RTPase activities would greatly strengthen the manuscript by making the measurement and analysis more physiologically relevant.

---

## [Author Response]

Essential revisions:The data in Figure 4 would be considerably strengthened by showing that interactions between S5:S11mut can be recovered by compensatory mutations in S5 and, vice versa, that S5mut:S11 interactions can be recovered by compensatory mutations in S11. Similar interaction 'rescue' experiments could be carried out for the S3:S11 and S6:S11 pairs presented in Figure 4—figure supplement 1.One major concern of this paper is the lack of in vivo validation for the observations made, and a lack of a well-reasoned model for how the formation of the higher-order ssRNA-oligomer allows itself for replication, duplex formation, and packaging. As noted above, we understand that in vivo validation is not practical at this point. Still, there are several questions (some of which could be addressed by in vitro experiments as well) such as for example 1) given that NSP2 strongly interacts with NSP5 in the viroplasm (and perhaps with other proteins including VP1 and VP2), how do these interactions affect NSP2-RNA interactions and RNA remodeling? Can the experiments be done in the presence of both NSP2 and NSP5? Do NTPase/RTPase activities of NSP2 play a role in RNA-RNA interactions or remodeling? This can be tested perhaps by using a mutant of NSP2 that lacks NTPase/RTPase activities? Addition of NSP5 and/or elucidation of potential roles of NTPase/RTPase activities would greatly strengthen the manuscript by making the measurement and analysis more physiologically relevant.

Here we have tried our best to respond to the wider concerns expressed by the referees.

In particular, we have addressed three general issues raised. These are:

1) Can the disrupted interactions between the mutated RNA sequences be ‘rescued’ by introducing compensatory, ‘trans-complementary’ mutations?

2) How does the viroplasm-forming RNA-binding protein NSP5 affect inter-segment RNA-RNA interactions alone or in the presence of NSP2?

3) Does the enzymatic activity (NTPase/RTPase) of NSP2 have an effect on the detected RNA-RNA interactions?

Regarding point 1, we have included results obtained with mutated RNAs S5 (S5mut) and a mutant of S11 RNA (comp_mutS11) showing that stable base-pairing between S5mut and the interacting S11 RNA mutant is restored. The mutated RNA sequences strongly interact with each other, consistent with the predicted stability of the ‘rescued’ RNA-RNA contacts between the two RNAs. These results, together with extensive characterisation of the stability of the formed duplex (Tm analysis presented in Figure 4) and mutagenesis analysis of both identified sites in S5 and S11 RNAs) further confirm that the identified RNA contacts are important for detected RNA-RNA interactions. We plan to ultimately test these sites in vivo, using the recently developed RG system by Kanai *et al.*, 2017. S11 RNA was chosen for mutagenesis analysis due to its small size and the availability of secondary structure probing data that could be used for introducing nucleotide substitutions whilst minimizing their impact on the secondary structure of the RNA.

Regarding point 2, we have expanded our studies to include RNA-binding protein NSP5 into RNA-RNA interaction assays. While both recombinantly expressed NSP2 and NSP5 bind ssRNAs, only NSP2 was capable of promoting the formation of RNA-RNA contacts between ssRNAs. Furthermore, addition of submicromolar amounts of NSP5 to the reaction involving ssRNAs S5 & S11, co-incubated with NSP2, resulted in less efficient formation of RNA-RNA contacts. Together these results strongly suggest that formation of RNA-RNA contacts requires NSP2, while NSP5 appears to inhibit this process while having other, not fully explored functions. In any case these observations are consistent with the previously reported NSP5 binding to the RNA-binding grooves of NSP2 octamers that competes with RNA binding by NSP2. We have also noted severe aggregation of the full-length NSP5 in the presence of NSP2, previously reported by Jiang *et al.,* 2006 which precluded further analysis of the RNA-RNA contact formation in the presence of both viroplasm-forming proteins NSP2 and NSP5.

Regarding point 3, we have included new results obtained with NSP2 in the presence of ATP. The novel data indicate that addition of ATP does not affect the ssRNA-binding activity of NSP2, nor does it affect the formation of the S5:S11 RNA-RNA complex. Furthermore, the S5:S11 RNA complex formation was significantly impaired when NSP2 was substituted with the DC-terminal NSP2 mutant that has reduced affinity for ssRNA, while still retaining its NTPase/RTPase catalytic residues. Furthermore, although not raised by the referees, we have investigated whether rotavirus group C NSP2 can facilitate the formation of RNA-RNA contacts between RNAs S5 and S11 of a group A rotavirus. Despite sharing similar tertiary and quaternary architectures and RTPase/NTPase enzymatic activities, rotavirus group C NSP2 was inefficient in stabilising the S5:S11 RNA-RNA contacts in rotavirus group A.

We feel that these additional results significantly enhance our initial conclusions and widen the appeal of the revised manuscript, and we hope that the Editor and the referees who felt the study was well designed, yet lacking in vivo validation for the observations made, are now more reassured.

Addressing these concerns and accommodating the new data generated have required us to rewrite sections throughout the manuscript (highlighted in yellow). We have also added a new figure (Figure 5) presenting a proposed model of NSP2-facilitated RNA-RNA assortment process, described in the Discussion.